# The Role of *Zwitterionic* Materials in the Fight against Proteins and Bacteria

**DOI:** 10.3390/medicines5040125

**Published:** 2018-11-22

**Authors:** Montserrat Colilla, Isabel Izquierdo-Barba, María Vallet-Regí

**Affiliations:** 1Departamento de Química en Ciencias Farmacéuticas, Facultad de Farmacia, Universidad Complutense de Madrid, Instituto de Investigación Sanitaria Hospital 12 de Octubre i + 12, Plaza Ramón y Cajal s/n, 28040 Madrid, Spain; mcolilla@ucm.es (M.C.); ibarba@ucm.es (I.I.-B.); 2Center on Bioengineering, Biomaterials and Nanomedicine (CIBER-BBN), 28040 Madrid, Spain

**Keywords:** biomaterials, *zwitterionic* surfaces, infection, bioceramics, bacterial inhibition

## Abstract

*Zwitterionization* of biomaterials has been heightened to a potent tool to develop biocompatible materials that are able to inhibit bacterial and non-specific proteins adhesion. This constitutes a major progress in the biomedical field. This manuscript overviews the main functionalization strategies that have been reported up to date to design and develop these advanced biomaterials. On this regard, the recent research efforts that were dedicated to provide their surface of *zwitterionic* nature are summarized by classifying biomaterials in two main groups. First, we centre on biomaterials in clinical use, concretely bioceramics, and metallic implants. Finally, we revise emerging nanostructured biomaterials, which are receiving growing attention due to their multifunctionality and versatility mainly in the local drug delivery and bone tissue regeneration scenarios.

## 1. Introduction

The increased usage of implantable medical devices is likely to result in a rise in the number of infections that are associated to these cases. Bacterial contamination during biomaterial implantation is often inevitable, provoking a battle between host cells and bacteria that may eventually cause the infection. This is a devastating complication that presents a heterogeneous clinical profile being considered as the most difficult infection disease to treat with serious clinical and socio-economic implications [1,2]. Biomaterials-associated infections generally include bacterial adhesion, colonization, and biofilm formation on the biomaterial surfaces. In general, these infections are mainly caused by different pathogens as *Staphylococcus epidermidis* and *Staphylococcus aureus* [3], although *Escherichia coli* and *Pseudomonas aeruginosa* are also present [4]. These bacterial colonizations result in an inflammatory reaction and are accompanied by significant morbidity and mortality rate. For all these reasons, there is an urgent need to develop biomaterials with improved properties that are able to provide solution to this serious clinical complication [4]. In this sense, preventive strategies that are aimed at inhibiting the first stages of any infective process constitute a powerful and promising alternative to tackle this issue [5].

Much research effort is being dedicated to develop new approaches to modify biomaterial surface to inhibit the bacterial adhesion. This could be an attractive alternative to antibiotics, which are associated with severe side effects, that moreover could create bacterial resistance. In this sense, it has recently been recognized that both nanotopography and chemical surfaces show an essential role in bacterial adhesion and biofilm formation [6]. In this sense, nanostructured surfaces currently represent a good alternative as bacterial-repelling surfaces [7,8]. These surfaces comprise a variety of nanotubes- or nanoparticle-based surfaces and nanostructured coatings [9], which create a superhydrophobic surface (also called leaves lotus effect) repelling the bacteria adhesion and compromising in most of the cases to the host cell-tissue integration [10].

Concerning chemical modification to create bacterial-repelling surfaces, *zwitterionization* has emerged as a revolutionary approach to provide biomaterials of high resistance to nonspecific protein adsorption, bacterial adhesion, and/or biofilm formation [11]. This strategy also allows for the preservation of the biomaterial biocompatibility in terms of host cell adhesion and colonization, cytotoxicity, and differentiation.

The aim of this review manuscript is to describe the different strategies developed so far for the *zwitterionization* of biomaterials. First, we focus on biomaterials in clinical use, concretely bioceramics and metallic implants. Later on, we revise the recent advances on nanostructured biomaterials, which have gained much attention by the scientific community, owing to their great potential and versatility in local drug delivery and bone tissue regeneration. To understand the processes underlying the different behaviors of *zwitterionic* surfaces by repelling bacteria, meanwhile allowing host cell colonization, an overview of the significance of the “race for the surface” concept between bacteria and eukaryotic cells is also given.

## 2. Tuning the Surface Properties of Biomaterials

The biocompatibility of the biomaterials is tightly related to the performance of cells when they came in contact and adhere to its surface. In this regard, the surface features of these materials, such as their topography chemistry or surface energy, become essential pillars in their biocompatibility [6]. Moreover, depending of the bioceramic functionality, the specific requirements in terms of cells/proteins adhesion are totally different. Therefore, in bone tissue regeneration applications, tuning the biomaterials surface to trigger bone bonding at the same time that inhibiting bacterial colonization constitutes an exciting challenge to achieve better clinical outcomes.

Bacteria exist in nature under two states: planktonic (free-floating bacteria) and biofilm (sessile microorganism communities). From all bacteria, just a small fraction (≈1%) exists in the free floating form, while around the 99% appear forming biofilms [12]. It is important to denote that the primary and harder barrier in treating *S. aureus* infections is often attributed to the formation of bacterial biofilm. In this sense, the biofilms are mainly built by big sessile bacteria communities that are embedded in a self-produced extracellular polymeric matrix or glycocalyx [13]. Within the biofilm, bacteria grow protected from environmental stress and resist attack by antibiotics, disinfectants, and the immune system. [14]. Biofilm formation involves a sequence of four steps, including: (i) adhesion, initial attachment of the bacteria to the surface of the tissue implant; (ii) growth, bacterial aggregation and accumulation in multiple bacterial layers; (iii) biofilm maturation; and, (iv) dispersion, detachment of bacteria from the biofilm and spreading to other places (asepsis state) [15]. 

In 1987, the orthopedic surgeon Anthony G. Gristina described the concept of “race for the surface” to predict the evolution of an implant in the specific relation to an infection process [16]. This concept contemplates “a race” between the eukaryotic cells (host cells) and the bacteria towards the implant surface, arguing that when the host cells colonize its surface, the probability of bacterial colonization is very low. However, when the implant is first colonized by the bacteria, it is irreversibly infected, not allowing the eukaryotic cell to colonize it. Thus, such a concept has stimulated technological and biomaterial progress while emphasizing the role of implant biocompatibility and tissue-integration. Thus, the great challenge is to design implants that make the race being won by eukaryotic cells, covering all biomaterial surfaces at the same time that inhibit bacterial colonization. However, the “race for the surface” concept has been criticized for its simplicity (simple rules) and the static conditions in which it is assessed, being notably useful for specific purposes in determining surface affinity to different species and the effects of coexistence [17]. Concerning this issue, the most destabilizing factor is the great survival rate of the bacteria, since they are able to adhere and survive on any surface [18,19].

The bacterial adhesion process can be divided into two main phases as reversible and irreversible stages, with the first phase being mechanically and biologically less stable than the second one [20,21]. The first phase, encompassing the bacteria adhesion and micro-colonies formation, is mainly governed by the electrostatic attraction forces between bacteria and surfaces (mediated by certain proteins like adhesins) and where the electrochemical nature of the biomaterial plays a major role [22,23]. On the contrary, the second phase is governed by molecular and cellular interactions that are closely related with the expression of specific gene clusters of the biofilm. They initiate the secretion of protective slime formed by mucopolysaccharide layer, which becomes extremely resistant to both host immune system and antibiotic diffusion [24]. It is important to remark that bacteria cannot initiate the biofilm-related phenotype before they firmly attach to the implant. Thus, the transition phase between reversible and irreversible processes of biofilm formation constitutes the last “window of opportunity” for clinically reasonable preventive treatments.

On the other hand, concerning the host site, the way that eukaryotic cells interact with the implant surface is through an interface that consists of discrete attachment protein points (integrins). These integrins interact with specific moieties of the extracellular matrix, such as RGD motifs [25], which contribute to bone regeneration and remodeling processes, being protected against bacterial colonization. However, neither osseointegration nor fibrous tissue encapsulation of an implant can eliminate the long-term survival of bacterial micro-colonies, which also contribute to a possible delayed infection. As a result, there is a strong need to design an intrinsic implant with antibacterial functionality that can overcome and kill these remnant bacteria [26].

Figure 1 shows the process of biomaterial surface colonization by bacteria starting from a reversible phase where individual floating microorganisms settle down by low stable adhesin-mediated non-specific interactions with the biomaterial. These first steps delimit a “window of opportunity” for almost all antibiofilm strategies, in which it is possible to inhibit the final biofilm formation and to reverse the final destiny of biomaterial for cell colonization. In this sense, if the host cell win the “race for the surface”, which attains irreversible attachments on the biomaterial surface, first, the presence of a continual cell layer makes it complicated for bacteria adhesion and biofilm formation.

## 3. Significance of *Zwitterionization* of Biomaterials

The development of *zwitterionic* bioceramics to inhibit unspecific protein adsorption was reported for the first time in 2010 by Colilla et al. [27]. Previously, Jiang and co-workers had described the *zwitterionization* to develop polymeric and metallic biomaterials fulfilling the ultralow-fouling criterion (<5 ng/cm^2^) [28,29]. *Zwitterionic* surfaces possess an equal number of both negatively and positively charged groups maintaining overall electrical neutrality, which depends on the pH of the environment [30]. Their non-fouling properties, as in the case of hydrophilic materials, are associated to the formation of a hydration layer on the surface of biomaterial, forming a physical and energetic barrier that hinders unspecific proteins adhesion. Recently, *zwitterionization* of bioceramics has emerged as a cutting-edge technology to confer surfaces not only of high resistance to non-specific protein adsorption, but also to bacterial adhesion and/or biofilm formation (Figure 2) [11,31]. However the main requisite of any biomaterial, biocompatibility, must be kept in mind, i.e., *zwitterionization* must prevent bacterial adhesion but allow for adequate host cell colonization. In vitro studies using osteoblastic-like cells revealed that they are able to appropriately adhere, colonize, and spread onto the surface of these *zwitterionic* bioceramics [31]. This different behaviour between prokaryotic and eukaryotic cells are caused by two main reasons [32]: i) the wall of the bacteria is formed by phospholipids layer, as eukaryotic cells, being much more rigid due to an external layer of peptidoglycan and ii) bacteria are much smaller in size (ca. 1 µm) than eukaryotic cells (ca. 50 µm). It has been proven that these both bacteria characteristics could be responsible of their capacity to discriminate differences in the biomaterials surfaces at the nanoscale level. As it has been above discussed in Section 2, since eukaryotic cells adhere to surfaces via integrins-mediated mechanisms, bacteria adhesion is mainly driven by electrostatic attractive forces that are mediated by adhesions. In this last case, the electrochemistry of the biomaterial surface is an essential factor governing their adhesion [22,23], which explains the different behaviour as compared to eukaryotic cells.

### 3.1. Chemical Strategies for the Zwitterionization of Biomaterials

In general terms, the *zwitterionization* of biomaterials involved in the functionalization of their surfaces at the atomic level [11] In the beginning, research efforts were focused on functionalizing with *zwitterionic* polymers bearing mixed positively and negatively charged moieties within the same chain and overall charge neutrality (Figure 3) [28,29,33,34,35]. There are three main methodologies to graft these polymers to the surface of biomaterials: (i) for functionalizing with poly(sulfobetaine) and polycarboxybetaine derivatives by surface-initiated atom transfer radical polymerization (SI-ATRP) [36,37,38,39]; (ii) for grafting sulfobetaine copolymers by surface reversible addition-fragmentation chain transfer (RAFT) through the polymerization method denoted as graft-from-surface”; and, (iii) more simple procedures through the polymerization method “graft-to-surface” [40]. In this method, polymers carrying adhesive moieties with strong surface affinity are synthesized and then grafted onto the surface through their adhesive moieties [41,42]. Furthermore, it is possible to confer biomaterials of *zwitterionic* nature by decorating their surface with low-molecular weight moieties bearing the same number of negative and positive charges (Figure 3). For instance, it is possible to functionalize with different amino acids, such as cysteine and lysine [43,44,45], sulfobetaine derivatives [46,47,48], or dopamine [49,50,51], exhibiting *zwitterionic* characteristics depending on the pH. Although the reported methods usually require several synthetic steps involving different intermediate products, they offer distinct advantages when compared to *zwitterionic* polymers, since they are usually associated to relatively more simple methods and lead to more biocompatible surfaces. A significant advance in the design and development of *zwitterionic* surfaces has consisted on the use of more direct and simple grafting methods by functionalization with organosilanes. In this case, the presence of hydroxyl (–OH) on the biomaterial surface helps to simultaneously attach two organosilanes bearing positive and negative charges, respectively. This strategy allows for tailoring the *zwitterionic* properties by adjusting the molar ratio of the two organosilanes that are used during the synthesis. This process can be accomplished using two different alternatives, the co-condensation and the post-synthesis route. In the case of co-condensation method, functionalization takes place at the same time that the biomaterial is being synthetized. For instance, *zwitterionic* mesoporous SBA-15 material containing NH_3_^+^/–COO^−^ groups was synthesized by adding two alkoxysilanes, aminopropyltrimethoxysilane (APTES) and carboxyethyl silanetriol sodium salt (CES) (Figure 3), together with the tetraethylorthosilicate (TEOS), as silica precursor, during the synthesis step [27]. Moreover, our research group has also reported the synthesis of *zwitterionic* SBA-15 by functionalization with (*N*-(2-aminoethyl)-3-aminopropyl-trimethoxysilane) (DAMO) alkoxysilane, which contains primary and secondary amine groups [52]. As it can be observed in Figure 3, its *zwitterionic* nature is provided by the presence of –NH_3_^+^/–SiO^−^ and >NH_2_^+^/–SiO^−^
*zwitterionic* pairs. On the other hand, the post-synthesis route relies on grafting the organosilanes to the biomaterials surface once they have been synthesized. Following this methodology, *zwitterionic* hydroxyapatite has been also been prepared by linking APTES and CES to the P–OH groups that are present on the surface of this biomaterial [53].

### 3.2. Zwitterionization of Biomaterials to Prevent Bacterial Infection

Currently, one of the major clinical challenges regarding the use of biomaterials is their custom-made design depending on the biomedical application [54,55]. Bioceramic implants for bone tissue regeneration, such as those that are based on calcium phosphates or bioactive glasses, can be manufactured as three dimensional (3D) scaffolds using rapid prototyping (RP) methods [54]. These scaffolds exhibit a high percentage of porosity and interconnectivity, and they are easily modulated with improved mechanical properties as compared to scaffolds that are fabricated by conventional methods [56]. Moreover, it is possible to combine nanostructural characteristics with micro-macro architecture for a fine-tuning of cellular behavior [57,58]. However, when facing the regeneration of large and critical bone defects, bioceramic implants are not suitable due to their intrinsic brittles [54]. It is feasible to manufacture metallic alloys using these RP methods, providing strong scaffolding to the bone regenerations purposes with porosities higher than 50% in volume and the rest being engaged by a metal skeleton [59,60]. In this regard, the milestone in bone tissue regeneration is to design these 3D scaffolds with surfaces that are capable of inhibiting and/or abolishing bacterial infection meanwhile allowing for osteoblast cells colonization. This surface would constitute a great technological advance to achieve better clinical outcomes. Currently, the scientific community is focussed on the design of 3D scaffolds that dynamically contribute to the regeneration process, stimulating the osteoconduction and angiogenesis at the same time than evade the bacterial infection [60,61,62,63]. However, the challenge is to provide 3D scaffolds of *zwitterionic* character. In this sense, 3D scaffolds based on pure nanocrystalline HA have been successfully constructed with a *zwitterionic* nature by the post-synthesis grafting of APTES and CES [53], which incorporates both –NH_3_^+^ and –COO^−^ groups on the surface, respectively (Figure 4, Left). To attain this goal, HA 3D scaffolds were first prepared by using the RP technique and then the resulting 3D-HA scaffolds were bifunctionalized by grafting both alkoxysilanes. Microbiological assays regarding bacterial adhesion using *Escherichia coli* (*E. coli*) showed a noticeable inhibition of 99% with respect to unmodified 3D-HA. The coexistence of –NH_3_^+^/–COO^−^ pairs onto 3D-HA scaffold avails its bacterial-repelling properties. At the same time, in vitro assays using HOS osteoblastic-like cells cultures demonstrated excellent biocompatibility as the cells were able to spread and colonize the entire scaffold surface. Scanning Electron Microscopy (SEM) micrographs show viable osteoblastic-like cells, exhibiting polygonal shapes with filopodia-like projections that are attached to the surfaces (Figure 4, Left). Moreover, the cell migration within the overall 3D-HA structure was demonstrated, showing the total colonization of 3D scaffold at different levels. This *zwitterionization* approach has been also applied onto Ti6Al4V 3D-scaffolds that were also fabricated by RP techniques [64]. In this case, to improve the functionalization capability of the metallic surface, it was previously coated by a HA layer using the dip-coating method. Once the HA coating was formed onto the Ti6Al4V 3D-scaffold, its surface was zwitterionized by the direct grafting of APTES and CES, following the same procedure as that reported for pure 3D-HA scaffolds [53]. Again, the presence of *zwitterionic* pairs inhibits *S. aureus* adhesion and biofilm formation, while permiting the osseointegration of this metallic implant, showing MC3T3-E1 preosteoblasts colonizing the entire scaffold surface. The obtained results indicate that the *zwitterionization* process does not affect the biocompatible properties of the metallic Ti6Al4V 3D-scaffolds, showing neither noticeable differences regarding cytotoxicity nor less proliferation as compared to bare 3D scaffold (Figure 4, right). Regarding the biofilm formation capability of these surfaces, we carried out confocal microscopy to study the biofilm formation after 24 h of incubation with *S. aureus* by using directly simultaneously acridine orange (green) and calcofluor (blue) fluorescent dyes, which label live bacteria and extracellular matrix of biofilms, respectively (Figure 4, Top). The obtained results clearly display the biofilm formation by the blue staining of a typical extracellular matrix covering the bacterial colonies with a thickness of 15 ± 3.3 μm on the Ti6Al4V scaffolds, while blue staining is absent in the Ti-Zwitter scaffolds, revealing the non-formation of the biofilm after 24 h of assay.

Concerning the regeneration process, we carried out a simple assay by confocal microscopy using preosteoblast MC3T3-E1 seeded on the surface of these scaffolds and incubated during 7 days. Both colonization and cell-morphology were studied by staining the cytoskeleton with Atto 565-conjugated phalloidin (red) and nuclei with DAPI (blue), as it can be observed in Figure 5 (bottom). In both casses, the cells display a high spreading grade with a well-built actin cytoskeleton and high level of colonization in all entire surface of both scaffolds. These results revealed that the *zwitterionization* process does not affect the healing process, showing the same behavior that unmodified Ti [64]. In the case of *zwitterionic* biomaterials currently under research, nanostructured bioceramics are receiving growing attention by the scientific community. Among these nanostructured biomaterials, silica-based mesoporous bioceramics are in the crest of the wave because of their exceptional features, such as high surface areas and pore volumes, tunable and narrow pore size distributions, and easy-to-functionalize surfaces [65,66,67]. Therefore they become excellent candidates to be provided of *zwitterionic* nature but also to host a great variety of antibiotics, allowing the combination of bacterial repellent and killing capabilities [68,69,70,71].

Figure 6 shows the performance of *zwitterionic* SBA-15 (SBA15-Zwitter) nanostructured bioceramic owning –NH_3_^+^/–SiO^−^ and >NH_2_^+^/–SiO^−^ pairs [52], provided by the co-condensation functionalization with the diamine alkoxysilane (DAMO) (Figure 3). This bioceramic exhibits the *zwitterionic* character at the physiological pH of 7.4, which constitutes a significant advance in this kind of materials for biomedical applications. In vitro bacterial adhesion tests using *S. aureus* strains reveal a great reduction of 99.9% with respect to unmodified SBA-15 (Figure 6). The intrinsic features of this nanostructured bioceramic permits the loading of the broad-spectrum antibiotic, cephalexin, showing a loading capability of around 13 mg∙g^−1^, together with a sustained drug release during more than 15 days. The synergistic combination of *zwitterionic* nature and antibiotic hosting capability opens up a new insight in the management of bone-associated infections. *Zwitterionization* has been also implemented on mesoporous bioactive glasses (MBG) having the SiO_2_–CaO–P_2_O_5_ composition [44]. These MBG are a type are nanostructured bioceramics analogous in composition to conventional bioglasses but exhibiting outstanding bioactive and cell response behaviors [69,72]. Thus, Sánchez-Salcedo et al. have recently reported the *zwitterionization* of MBG by tethering lysine (MBG-Lys). In vitro bacterial adhesion assays with *S. aureus* proved a reduction up to 99.9% as compared to unmodified MBG. Moreover, MBG-Lys are cytocompatible, as demonstrated by in vitro studies carried out with MC3T3-E1 preosteoblasts cultures, which increases the potential application of these nanostructured bioceramics in bone tissue regeneration.

## 4. Conclusions

*Zwitterionization* is emerging as a powerful strategy to design advanced biomaterials for the management of infection that is envisioned to result in better clinical outcomes. The possibility to easily functionalize biomaterials surface at the atomic and nanoscale levels to prevent the non-specific protein and bacterial adhesion opens up many paths to tackle severe clinical concerns. Indeed, *zwitterionic* biomaterials have shown an opposite behaviour by inhibiting bacterial adhesion, while allowing host cells adhesion and colonization of the surface. This fact constitutes the cornerstone in their potential clinical application and much research effort is committed to translate these significant advances from bench to bedside. However, there are certain challenges that these biomaterials have to face for their clinical stage of development. These challenges include: (i) preservation of biocompatibility, (ii) adequate pharmacokinetics through the local administration of antimicrobial agents, and (iii) bone healing capacity, which guarantee success in bone regeneration.

## Figures and Tables

**Figure 1 medicines-05-00125-f001:**
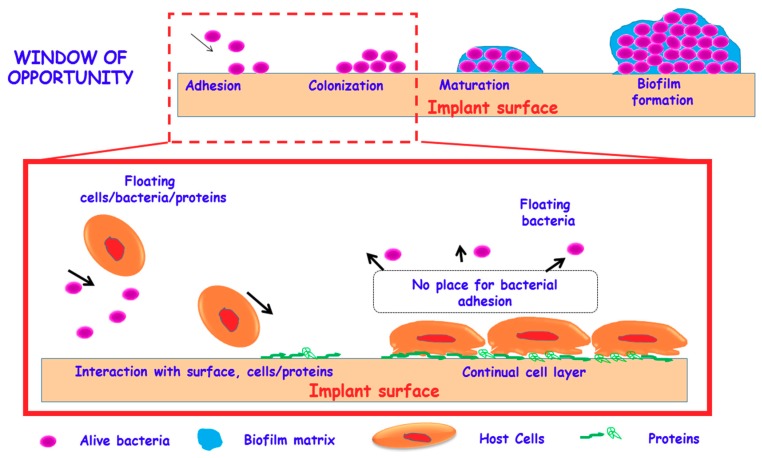
Concept of “race for the surface” and significance of “window of opportunity” in the development of implants capable to be colonized by host cells, while impede bacteria adhesion and the formation of the biofilm.

**Figure 2 medicines-05-00125-f002:**
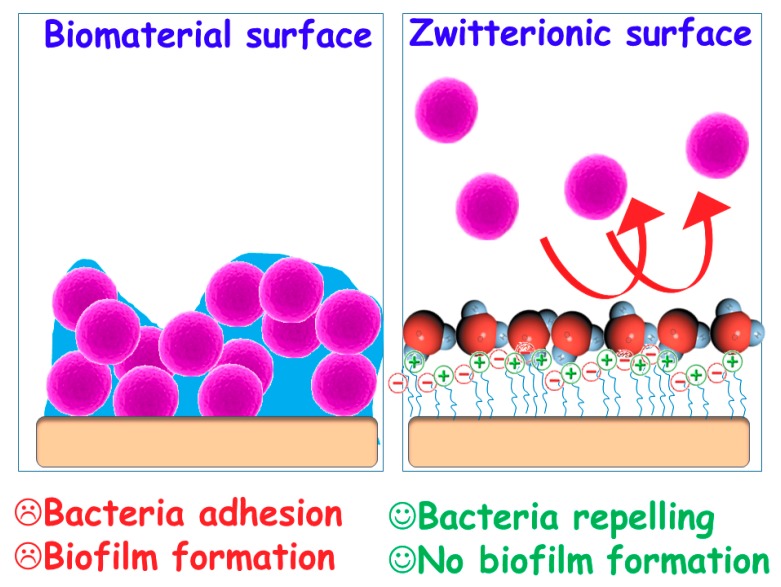
Schematic illustration of the different behavior of conventional biomaterial surfaces vs biomaterials *zwitterionic* surfaces against bacterial colonization. As bacteria get close to the surface of conventional biomaterials, they are able to adhere, colonize and forming a biofilm, which is one of the major concerns in biomaterials associated infections. Oppositely, *zwitterionic* surfaces provide biomaterials of bacterial-repelling properties, thus inhibiting the subsequent biofilm formation, which constitutes a promising alternative in the biomaterials scenario to prevent bacterial infection.

**Figure 3 medicines-05-00125-f003:**
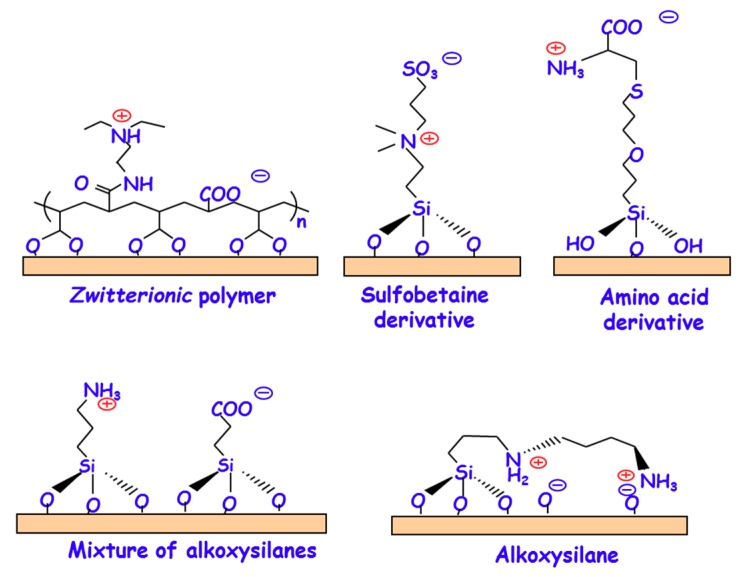
Representative examples of the different chemical strategies developed so far for the *zwitterionization* of biomaterials: Grafting of *zwitterionic* polymers [e.g., 3-(diethylamino)propylamine (DEAPA) coupled to poly(acrylic acid) (PAA)]; Sulfobetaine siloxane derivatives; Amino acid derivatives (e.g., cysteine); Mixture of alkoxysilanes [e.g., 3-aminopropyltrimethoxysilane (APTES) and carboxyethylsilanetriol sodium salt (CES)]; Alkoxysilane [*N*-(2-aminoethyl)-3-aminopropyl-trimethoxysilane) (DAMO)].

**Figure 4 medicines-05-00125-f004:**
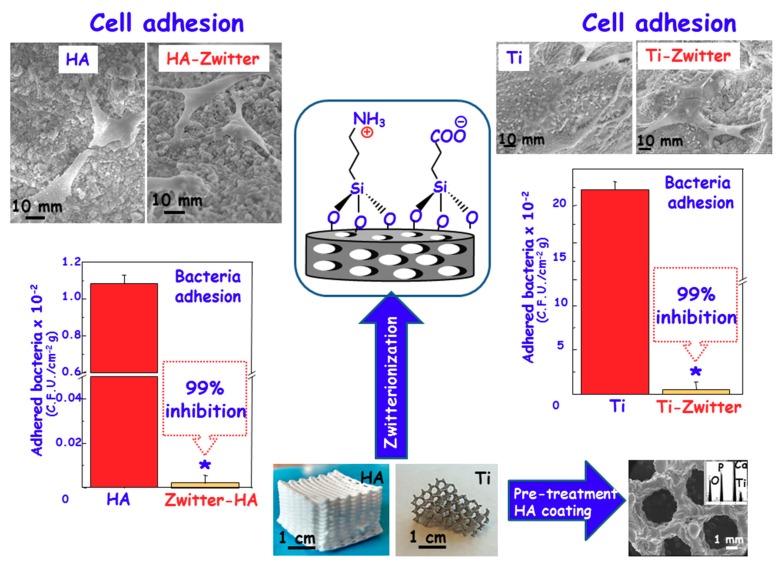
*Zwitterionization* of different biomaterials in clinical use by grafting of carboxyethylsilanotriol sodium salt (CES) and aminopropyltriethoxysilane (APTES) onto the surface of these biomaterials. Left: Pure HA three-dimensional (3D) Rapid Prototyping (RP) scaffolds. Right: Electron Beam Melting (EBM) Ti6Al4V 3D scaffolds.

**Figure 5 medicines-05-00125-f005:**
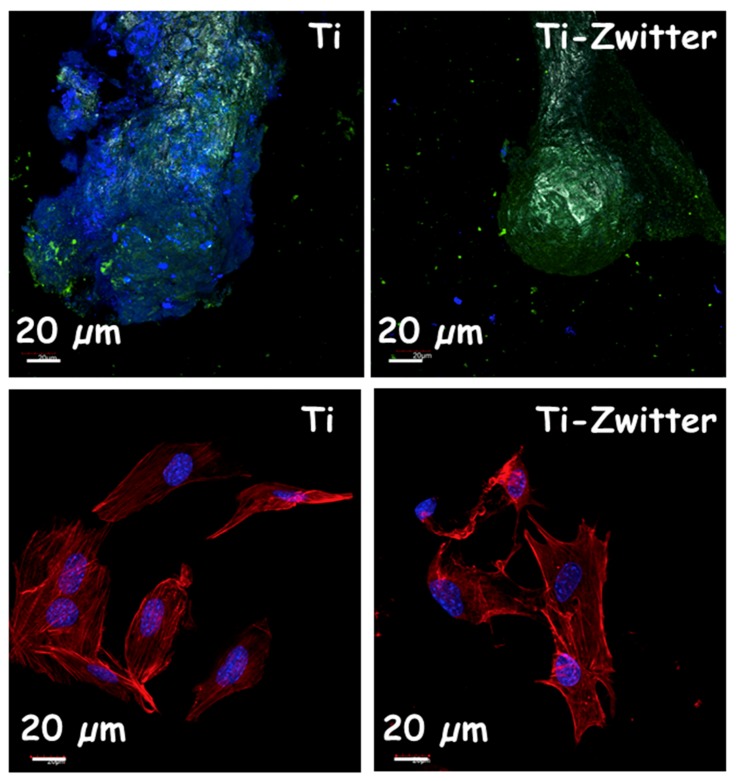
Confocal microscopy images showing of EBM-Ti6Al4V 3D scaffolds before (Ti) and after being *zwitterionized* (Ti-Zwitter). In vitro behavior of the biomaterials after being incubated during 24 h in *S. aureus* bacteria (Top) and during 4 days in MC3T3-E1 preosteoblast cells (Bottom). The results reveal the biofilm formation on the Ti surface, appearing the blue coating corresponding to the polysaccharide matrix (calcofluor), whereas this coating is not observed on the surface of Ti-zwitter surface, which confirms the antibiofilm formation preventing capability of this material. Both Ti and Ti-zwitter surfaces undergo an appropriate preosteoblastic colonization and spreading in the entire surface, with the nuclei stained in blue (DAPI) and the cytoskeleton stained in red (phalloidin).

**Figure 6 medicines-05-00125-f006:**
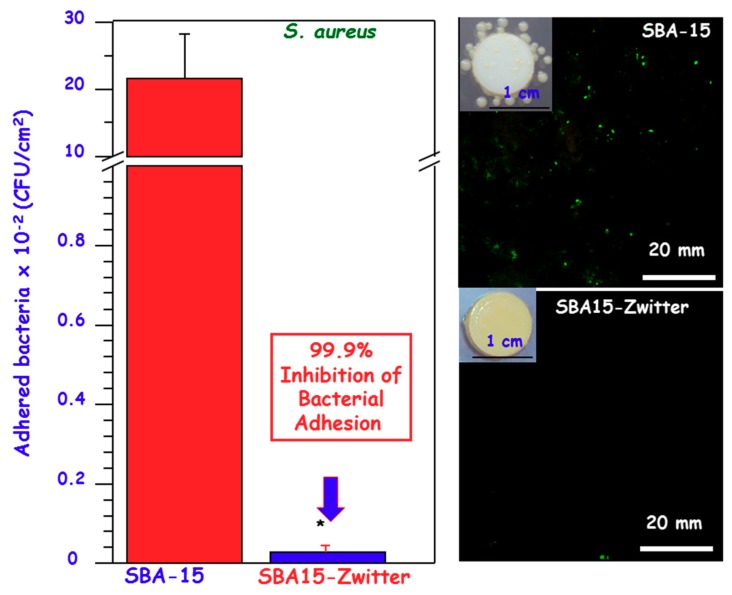
Schematic depiction of pure silica SBA-15 and *zwitterionic* SBA-15 (SBA-Zwitter) nanostructured materials. SBA-Zwitter was prepared following the co-condensation route in the presence of DAMO. Counting of colony forming units of *S. aureus* after 90 min of culture onto SBA-15 and SBA15-Zwitter surfaces. Statistical significance: * *p* < 0.01. Confocal microscopy images of *S. aureus* adhered onto SBA-15 and SBA15-Zwitter surfaces after staining with Baclight^®^ KitTM.

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
