# Peer review of "The Role of Zwitterionic Materials in the Fight against Proteins and Bacteria"

_medicines, 2018, doi:10.3390/medicines5040125_

Round 1

Reviewer 1 Report

I find the subject of the review interesting, it is well organized and I recommend publication of this review after minor revisions.

Please consider a slight modification of the title: in my opinion, it is not correct English.

Please change the title of section 2 so that it would describe the contents of this section (i.e. bacterial adhesion and biofilm formation)

line 23: the increased usage of implantable devices in medicine is likely to result in a rise...

lines 29/30: delete 'rendering the infection' or rephrase the sentence

lines 30-32: is it possible to provide the reference? Are there other microorganisms than S. aureus, such as Pseudomonas aeruginosa, that cause infections associated with biomaterials

line 31: result (not results)

lines 36/7: please, rephrase 'antibacterial adhesion capability' sounds strange

line 82: please, rephrase

line 104: please, rephrase

line 110: survival, not survivorship

line 117: please, rephrase

line 131: is this overall charge affected by pH?

fig 2: Non-biofilm formation is not correct. Did you mean no biofilm formation/no formation of biofilm?

line 171/171: For instance, not for example

line 215: what does it mean 'the Holy Grail'? did you mean the milestone?

lines 226-250: S. aureus, E. coli- italics

lines 281-294: italics : in vitro, S aureus

Author Response

Reviewer #1

Comments and Suggestions for Authors

I find the subject of the review interesting, it is well organized and I recommend publication of this review after minor revisions

Please consider a slight modification of the title: in my opinion, it is not correct English

We really appreciate the referee comment and the title of our manuscript has been changed as follows:

“The role of zwitterionic materials in the fight against proteins and bacteria”

Please change the title of section 2 so that it would describe the contents of this section (i.e. bacterial adhesion and biofilm formation)

We thank this suggestion, however in this section is also included the protein adsorption which is in relation with the posterior host cell and bacterial adhesion. Thus this section has been modified as follows:

“2. Tuning the surface properties of biomaterials to enhance the biocompatibility

line 23: the increased usage of implantable devices in medicine is likely to result in a rise...

Done

lines 29/30: delete 'rendering the infection' or rephrase the sentence

Done, we have deleted the term “rendering the infection”

lines 30-32: is it possible to provide the reference? Are there other microorganisms than S. aureus, such as Pseudomonas aeruginosa, that cause infections associated with biomaterials

The referee totally agree and the sentence has been change as follows:

Line 28: “Biomaterials-associated infections generally include bacterial adhesion, colonization and biofilm formation on the biomaterial surfaces. In general, these infections are mainly caused by different pathogens as Staphylococcus epidermidis and Staphylococcus aureus [3], although Escherichia coli and Pseudomonas aeruginosa are also present [4]. These bacterial colonizations result in an inflammatory reaction and are accompanied by significant morbidity and mortality rate.”

Moreover, a new reference has been provided (reference #3):

“Campoccia, D.; Montanaro, L.; Arciola C.R.; The significance of infection related to orthopedic devices and issues of antibiotic resistance. Biomaterials 2006, 27, 2331.”

line 31: result (not results)

Done

lines 36/7: please, rephrase 'antibacterial adhesion capability' sounds strange

We totally agree with the referee comment and the sentence “antibacterial adhesion capability” has been modified as follows:

Line 38“….to inhibit the bacterial adhesion.”

line 82: please, rephrase

We rephrase the sentence as follows:

Line 78: “In 1987, the orthopedic surgeon Anthony G. Gristina described the concept of "race for the surface" to predict the evolution of an implant in the specific relation to an infection process [16]. This concept contemplates "a race" between the eukaryotic cells (host cells) and the bacteria towards the implant surface, arguing that when the host cells colonize its surface, the probability of bacterial colonization is very low. However, when the implant is first colonized by the bacteria, it is irreversibly infected, not allowing the eukaryotic cell to colonize it.”

line 104: please, rephrase

We rephrase the sentence as follows:

Line 97: “On the contrary, the second phase is governed by molecular and cellular interactions closely related with expression of specific gene clusters of the biofilm. They initiate the secretion of an protective slime formed by mucopolysaccharide layer, which becomes extremely resistant to both host immune system and antibiotic diffusion [24].”

line 110: survival, not survivorship

Done

line 117: please, rephrase

We rephrase the sentence as follows:

Line 114: “These first steps delimit a “window of opportunity” for almost all antibiofilm strategies, in which is possible to inhibit the final biofilm formation and to reverse the final destiny of biomaterial for cell colonization. In this sense, if the host cell win the “race for the surface”, which attain …”

line 131: is this overall charge affected by pH?

Yes, this overall charge depends on the pH of the environment. In order to clarify this, this sentence has been included. 

fig 2: Non-biofilm formation is not correct. Did you mean no biofilm formation/no formation of biofilm?

The figure has been corrected.

line 171/171: For instance, not for example

Done

line 215: what does it mean 'the Holy Grail'? did you mean the milestone?

Done

lines 226-250: S. aureus, E. coli- italics

Done

lines 281-294: italics : in vitro, S aureus

Done

Reviewer 2 Report

In this manuscript, Colilla et al. summarized recent advances in the development of zwitterionic biomaterials and their applications in preventing bacterial infection and protein adsorption. The manuscript is well-organized and developed, I suggest this manuscript could be considered for publication in Medicines.

1. The authors talked more about the biomaterials against bacterial than proteins, the recent advance of biomaterials against proteins should be further discussed.

2. The language of the manuscript needs to be further polished. The current version contains many grammar errors.

3. In the conclusion part, it is better to discuss challenges and future directions for zwitterionic biomaterial development.

Author Response

Reviewer #2

Comments and Suggestions for Authors

In this manuscript, Colilla et al. summarized recent advances in the development of zwitterionic biomaterials and their applications in preventing bacterial infection and protein adsorption. The manuscript is well-organized and developed, I suggest this manuscript could be considered for publication in Medicines.

We really appreciate the referee comments.  

1. The authors talked more about the biomaterials against bacterial than proteins, the recent advance of biomaterials against proteins should be further discussed.

This manuscript is focused mainly in the design of new biomaterials with antibacterial functionality that can overcome and kill these remnant bacteria meanwhile contribute to bone tissue regeneration with appropriated osteoblast cells colonization. The initial non-specific protein adsorption is related to the final destination of the implant. That is, colonized by the bacteria or by the host cell. In this sense, section #2 of our manuscript describes these phenomena. To clarify this, this section has been slightly modified in order to understand these processes.

2. The language of the manuscript needs to be further polished. The current version contains many grammar errors.

Thank you very for the comments, we really appreciate it. Thus, the language of the manuscript has been polished and several grammar errors have been also corrected.

3. In the conclusion part, it is better to discuss challenges and future directions for zwitterionic biomaterial development.

We agree to the referee comment and we have included the following sentence in conclusion section:

Line 316: “However, there are certain challenges that these biomaterials have to face for their clinical stage of development. These challenges include: (i) preservation of biocompatibility, (ii) adequate pharmacokinetics through the local administration of antimicrobial agents and (iii) bone healing capacity, which guarantee success in bone regeneration.”
